# Community Group Model Building as a Method for Engaging Participants and Mobilising Action in Public Health

**DOI:** 10.3390/ijerph17103457

**Published:** 2020-05-15

**Authors:** Sarah Gerritsen, Sophia Harré, David Rees, Ana Renker-Darby, Ann E. Bartos, Wilma E. Waterlander, Boyd Swinburn

**Affiliations:** 1School of Population Health, University of Auckland, Auckland 1142, New Zealand; sophia.harre@auckland.ac.nz (S.H.); aren830@aucklanduni.ac.nz (A.R.-D.); boyd.swinburn@auckland.ac.nz (B.S.); 2Synergia Consulting Ltd, Auckland 1011, New Zealand; david.rees@synergia.co.nz; 3School of Environment, University of Auckland, Auckland 1142, New Zealand; a.bartos@auckland.ac.nz; 4Department of Public Health, Amsterdam UMC, University of Amsterdam, 1105 Amsterdam, The Netherlands; w.e.waterlander@amc.uva.nl

**Keywords:** systems analysis, public health, poverty, vegetables, system dynamics, group model building, participatory research, children, qualitative methods

## Abstract

Group model building (GMB) is a qualitative method aimed at engaging stakeholders to collectively consider the causes of complex problems. Tackling inequities in community nutrition is one such complex problem, as the causes are driven by a variety of interactions between individual factors, social structures, local environments and the global food system. This methods paper describes a GMB process that utilises three system mapping tools in a study with members of a multicultural, low-income community to explore declining fruit and vegetable intake in children. The tools were: (1) graphs over time, which captures the community’s understanding of an issue; (2) cognitive mapping, which enables participants to think systemically about the causes and consequences of the issue; (3) causal loop diagrams, which describe feedback loops that reinforce the issue and identify potential actions. Cognitive mapping, a tool not usually associated with GMB, was added to the research process to support the gradual development of participants’ thinking and develops the skills needed to tackle an issue from a systems perspective. We evaluate the benefits and impact of these three tools, particularly in engaging participants and increasing understanding of systems thinking in order to develop and mobilise action. The tools could be adapted for use in other community-based research projects. Key learnings were the value of genuine partnership with a local organisation for longevity of the project, recruitment of key decisionmakers from the community early in the process, and allowing time to create sustainable change.

## 1. Introduction

Tackling a complex public health issue such as poor nutrition is a challenging task. The causes of poor nutrition are complex and multiple [1], with prevalence driven in part by the interaction between the global food system and local environments [2]. There is no single solution, and improving population diet requires action across many individuals, groups and communities [3,4]. Poor nutrition is not just a health issue but is embedded in a social, political and cultural context; for example, in the way food is grown (i.e., agricultural subsidies), produced and marketed (e.g., influence of large multinational companies) and consumed (e.g., cultural traditions around food) [5]. It is therefore essential to consider the system within which poor nutrition sits when conducting research and implementing policies.

System dynamics is a computer-aided approach to understanding such complex systems. While system dynamics has always had a tradition of involving people to provide input to develop simulation models, it was only in the mid-1990s that leaders in the field recognised the value of involving people directly in the model building process [6,7]. Incorporating stakeholders such as community members in the process resulted in models that were more relevant to the issues being tackled provided more shared insights amongst those who were charged with tackling the issues and, through the participation, provided more motivation for implementing the results [6,8]. In New Zealand, one of the authors utilised a system dynamics simulation model to engage a wide range of stakeholders in exploring a range of options for cardiovascular disease interventions [9]. The group model building (GMB) process that emerged out of this study emphasised that engaging key stakeholders in a model building process is important and an effective way to create collaborative models [10,11].

GMB involves community members using informal causal loop diagrams (CLDs) to explore the multiple, interacting feedback loops operating in the system of interest [7,8]. CLDs provide a visual image of an issue, allowing participants to see both their contribution to the model building effort as well as the areas where they can contribute change to the system. CLDs are appropriate for use by community groups, as they are easier to build and understand and have fewer data and resource requirements than quantitative simulation models. Furthermore, community groups can take over ownership and ongoing development of the model after training, without having to acquire the specialist software, or skills needed to develop and use quantitative simulation models.

The use of GMB in public health research is still relatively new. Several studies have successfully explored childhood obesity with GMB [3,12,13,14], and a recent study in Baltimore used GMB to explore suboptimal availability of healthy food in a low-income urban community [15]. A unique aspect of GMB has been its effectiveness within multicultural communities, and its ability to accommodate and acknowledge indigenous worldviews [16].

This methods paper describes a GMB process used in a study that engaged members of a diverse urban community to explore declining fruit and vegetable intake among children—its causes, consequences and potential interventions to improve public health. We evaluate the benefits and impact of the three tools used in the GMB process—graphs over time (also known as ‘reference behaviour patterns’), cognitive mapping, and causal loop diagrams—in order to inform future researchers wishing to use this method. A paper describing the findings from the study has been published elsewhere [17].

## 2. Materials and Methods

### 2.1. Community Partnership

Academic researchers from the University of Auckland, supported by a consultant with expertise in system dynamics, partnered with Healthy Families Waitākere (HFW). HFW is a community prevention-based programme funded by the Ministry of Health, working to improve health outcomes. A key focus of their work had been improving nutrition in their local communities and a partnership with the University supported their work in this area. Over three consecutive weeks, the organisations co-facilitated three GMB workshops with members of a city-fringe suburban community in West Auckland, New Zealand. HFW has extensive knowledge of, and networks within, West Auckland communities and expressed commitment to catalysing action on this issue beyond the end of the project. The project’s research assistant was based at the HFW office. A low-income, culturally-diverse suburb, which had 11 schools and a high proportion of young families in the population, was chosen as the location for the research, and the workshops were held at a local high school.

### 2.2. Participant Recruitment Process

Ethical approval for the study was obtained from the University of Auckland Human Participants Ethics Committee (ref 020684). The participant recruitment process was led by HFW and supported by the research assistant. A diverse range of participants representing different aspects of the community were recruited, including local food retailers; school principals, teachers and students; parents; and local health promoters. Participants were given a $100 voucher at the third workshop to thank them for their participation in the research.

### 2.3. GMB Workshop Process

The goal of the workshops was to assist the community to develop a richer understanding of food and vegetable (FV) intake in their community and to illustrate that understanding in a causal map describing the drivers of FV intake in their community, as well as possible points for intervention. A total of three, two-hour workshops were held over three consecutive weeks. The workshops followed a structured format and utilised several GMB scripts [18,19] available from Scriptapedia [20] that were modified slightly to suit the local context. 

All three workshops had a similar structure and each exercise built on the skills developed in the previous workshop. The workshops followed Māori tikanga (protocol); prior to the start of each session, there was a karakia (Māori prayer or blessing) and a shared meal and activity designed to assist with whakawhanaungatanga (building relationships and getting to know one another). Each workshop closed with a karakia. Members of the research team were assigned different roles during the workshops (Table 1). 

Participants worked in groups of 3–5 per table and a member of the research team sat at each table to act as a facilitator. New groups were made at the start of each workshop. Groups presented their work and their findings were discussed frequently throughout the workshops/before moving onto the next task. Detailed run sheets and ‘cheat sheets’ were provided to each table facilitator before the workshops. The cheat sheets contained group tasks, including the purpose of each task, the process, prompt questions for facilitators to use if participants were stuck, and a reminder to capture the conversations on the worksheets.

The research team held an inter-workshop meeting the day following each workshop to ensure the key topics were captured and correctly interpreted. The team was able to reflect on the process and made necessary adjustments for the following workshop.

### 2.4. GMB Tools

The three tools which formed the basis of the workshops were (1) graphs over time, (2) cognitive mapping and (3) causal loop diagrams. We now outline the three tools to provide a blueprint for future research.

#### 2.4.1. Graphs over Time 

Graphs over time enable participants to share their understanding of an issue and what has changed in their community over a certain time period. Participants were introduced to an empty graph with time on the X-axis (with a vertical line for the present time) and a variable on the Y-axis [20]. To define the variables in our study example, participants focused on the question; “What are the factors that have affected or influenced children in (your community) eating fruit and vegetables?” They were asked to draw on the empty graphs the historical pattern and two future pathways they predicted would occur if current trends continued or if intervention occurred (Figure 1).

#### 2.4.2. Cognitive Mapping

Cognitive mapping [21,22] is a visual tool that teases out participants’ understanding of the causes of an issue, exploring their causal reasoning. Cognitive mapping also elicits an understanding of why the issue is important to them by mapping out their perspectives of the consequences of addressing the issue, successfully or not. A template was provided (Figure 2) that participants completed.

#### 2.4.3. Causal Loop Diagrams

Causal loop diagrams highlight the dynamic nature of an issue and the presence of feedback in systems. The aim of this task was to provide participants with an understanding of why feedback is important in a system and ground the teaching examples in work they had already done. To do this, simple feedback loops were developed out of the participants’ causes and consequences tables (Figure 2). Some research participants can find CLDs challenging to build, read and use. Therefore, we introduced CLDs only after introducing causality through the use of cognitive mapping, which naturally led to the introduction of feedback loops. Feedback loops are a basic operating unit of systems represented in the CLDs [23]. It is common for research participants to both misunderstand and misinterpret feedback [24]. 

The research team consolidated and merged the groups’ ideas from each table into one collective CLD during a meeting between workshops. At the next workshop, each group was given a copy of the collective CLD and a list of variables from the graphs over time that were not included in their original CLDs. In groups, participants were asked to critique the collective CLD by adding, deleting and modifying structures in the map.

The next stage was to explore potential interventions to the barriers identified in the CLDs. Participants were asked to explore potential actions which could be taken, using the following questions for guidance:What variables could you increase or decrease?How could you impact connections: strengthen, or weaken a connection, speed it up or slow it down, add or delete connections?How could you impact the ‘rules’ that govern the system or the goals that it is trying to achieve?

In their groups, participants wrote potential actions on post-it notes and placed these on the CLD where they felt they would fit (Figure 3). They then selected the ‘top five’ actions that their table would like to see progressed.

#### 2.4.4. Post-Workshop Activity

Following analysis of the workshop outputs, the research team handed over the project to HFW to work with the community to catalyse action. Approximately 12 months after the GMB workshops, two members of the research team met with staff from HFW to discuss what had happened in the interim with the purpose of evaluating the benefits and impact of the GMB process. The meeting was audio recorded to provide quotes for inclusion in this manuscript. 

## 3. Results

The majority of participants were recruited through HFW’s existing networks, following an invitation from a HFW staff member whom they already had a relationship with. Once invited by HFW, a number of participants took part in the recruitment process. This approach included participants recruiting through their own networks (snowballing), resulting in 17 participants (14 of whom attended all three workshops). 

The workshops were held in a local school, on Tuesday evenings, for three consecutive weeks. During the first activity, graphs over time, participants worked in small groups of three to five people per table, and participants developed a number of their own graphs. The graphs were seen as expressions of community understanding, reflecting what the community saw as important factors affecting children’s intake of fruit and vegetables in their community. The activity served two purposes. Firstly, they enabled the research team to identify what the community thought were key factors affecting the uptake of fruit and vegetables and how those factors had evolved over the last few years. Secondly, the exercise helped to move participants toward a more systemic and dynamic way of thinking about such issues by getting them to think about change over time and not just about the situation as it was now [17,18]. In addition to encouraging participants to think about key variables changing over time, the graphs illustrate that the future has more than one possible pathway.

For the second activity, cognitive mapping, participants completed the causes and consequences template (Figure 2). In the “causes” boxes, they provided the key factors and causal relationships that they believed were affecting the consumption of fruit and vegetables by children (Figure 4). The “consequences” boxes provided information on why they saw the consumption of fruit and vegetable by children as important (Figure 4). The consequences listed were later used to focus the interventions and provide motivation for work beyond the completion of the workshops. Four of these templates were completed in workshop one. Discussions within and between groups allowed participants to share ideas and extend their maps, increasing awareness of the multiple causes and consequences. Thorough this process, participants developed a shared understanding of the complex nature of the food and nutrition environment, to the point where they could describe it in their own words.

The third activity took place at the second workshop, following an introduction on feedback and CLDs, using a worked example. Participants wrote the variables from their causes and consequences tables and graphs over time on post-it notes and used these to develop CLDs in groups of three to five with a facilitator at each table (Figure 5).

The main outputs from the workshops were the final collective CLD and the list of actions proposed by workshop participants (described in detail in the results paper [17]).

The feedback from participants to HFW on the use of these tools was positive. The focus on building the participants’ capability one step at a time ensured that the participants understood each stage of the GMB process. 

“It was the third workshop that the stories really started coming out, sharing their struggles about not having the time or the money, working two jobs.”

Three months after the final workshop, HFW called a meeting to identify some priorities for action. Three workshop participants attended this meeting, and it was unsuccessful at generating immediate action. In 2019, another meeting with approximately 20 attendees (including 3 participants from the original workshops) was held which focused on prioritising action and agreement was reached on actions to push forward to increase fruit and vegetable access in the local schools. Attempts to gain the support of a local supermarket to regularly donate fruit were unsuccessful, and therefore they sourced fruit from the local community to trial a ‘fruit station’ at a local high school (roll: 750). The idea showed promise, as students consumed the majority of the produce. However, it was acknowledged that such as intervention on its own is unlikely to make a significant contribution to increasing children’s overall FV intake and relying on community donations may not be a sustainable approach (personal communication, HFW). 

HFW’s reflections on the GMB process were that, in order to meet the research goal of catalysing community actions, the action phase needed to be explicitly designed into the workshop process. A HFW team member explained:

“If we were to do this again with the intention to mobilise the community, then we need to have some key players, a prominent leader (like the local MP) to lead it and be there right from the start and that might have got us to that mobilisation phase earlier... We are doing it now in an ad hoc way.”

It was important to recruit not just community members to share their lived experiences and understanding of the issue, but also community leaders to enable action.

“...who has the power? We had people experiencing the impact of poor access to fruit and veg, but the people who have the power to change things we needed to think about them from the start.”

A year after the completion of the GMB workshops, HFW are continuing to use the GMB outputs to enroll support for community-wide responses to the challenge of low fruit and vegetable consumption among children in their local community. In addition, the outputs from the three workshops have been presented to their local Member of Parliament, community board, and Board of Trustees at a local high school, to push for the development of a coordinated community strategy. The response to the presentations and meetings held by HFW has been positive. As they described it:

“They were really positive about the whole kaupapa (philosophy/purpose of the research), the process, how we went about it and who turned up and participated.” 

Advocacy work is continuing and, as described to the research team, the workshop outputs and process were ensuring that the ‘voice of the community’ is strong and clear in advocating for change.

“We did get traction this year (following the workshops)… brokering partnerships, and then leveraging from the research—discussed with a local MP to amplify and get some traction. We shared the draft manuscript (with the results from the GMB), and that bought in good engagement from her, but also other MPs as well. We prepared a walk-through and presented that to the local community board and two Board of Trustees meetings… It was honouring what the community was saying.”

## 4. Discussion

### 4.1. Commentary on the Use of the Three Tools

The tools chosen, and the workshop structure used, were designed to support a development in the thinking of participants. They assisted participants in shifting their thinking from isolated factors, such as the marketing of fast food, to seeing these factors as evolving over time, driven by causal connections that were sitting within positive and negative feedback loops. It is important to recognise that this process can be challenging. For many people, this constitutes a significant shift in thinking patterns and requires the use of tools that help them describe and frame issues, elicit knowledge and beliefs from others, while mapping the system structure that emerges from this knowledge [25]. Because of the challenges in helping people make this shift in thinking, the workshops were designed to take participants through a step-by-step process that introduced key skills in a sequential and cumulative manner. The use of cognitive mapping to introduce ideas of causality before introducing the concept of feedback facilitated this shift in thinking.

Graphs over time and the creation of CLDs are both standard GMB tools. The novel tool added to our workshop format was cognitive mapping, which provided a bridge between graphs over time and the creation of CLDs. While graphs over time engaged participants in thinking about change over time rather than points in time, CLDs focused on the feedback dynamics involved in that change. However, CLDs require people to not only come to grips with new concepts, such as causality and feedback, but also to learn the ‘rules’ and syntax associated with the tool. Cognitive mapping introduces participants to the concept of causality without them having to learn about feedback or the specific syntax of CLDs. The process allowed community members to tell their own stories and develop a better understanding of children’s fruit and vegetable intake, its causes and its consequences, and we would therefore recommend the use of cognitive mapping with the causes and consequences template for future GMB research.

### 4.2. Partnering with a Local Community

Research in multicultural, diverse communities is always challenging due to potential power imbalances, misunderstandings about motives, and difficulty recruiting participants who may have more pressing commitments [26]. The priority for the research team in this study was to be respectful of the participants and create an environment where they felt comfortable and confident when communicating and sharing their ideas. 

Partnering with HFW was essential to the success of the project due to their extensive knowledge of the local area and their relationships within the community. The process strengthened existing relationships within the community and the training provided by the system dynamics consultant enhanced GMB skills for both partners. The intention from the start of the project was that HFW would take the proposed actions forward alongside the community, and so the outputs would have “a home” and would continue to be developed.

### 4.3. Engaging Participants in the GMB Process

Similar to other public health researchers that have used GMB [15], this study has demonstrated that GMB tools can effectively engage participants and develop their understanding of system dynamics. The collective process of GMB allowed participants to explore issues faced by their community and elicited individual experiences. 

Other aspects that facilitated engagement were following Māori cultural process for building relationships (allowing 30 minutes prior to each workshop to share food and have an ‘icebreaker’ activity), and having individual table facilitators to ensure that each member of the group was given the opportunity to contribute and that each member of the group understood the process.

### 4.4. Creating Change

If the goal is to produce models that can be used to generate action—as intended by the researchers in our study, and described by Hovmand [27]—more work needs to be done to ensure that the implementation aspects are given due consideration in the design phase of GMB research [27]. Hovmand (2012) admits that “the delay between when they participated and when their initiatives appear can be on the order of several years or more” from the GMB process [7]. 

To proactively increase the chance of successful action resulting from the research, a subsequent co-design process with participants to develop potential interventions could be built into GMB projects from the beginning. Anselma et al. (2019) used a 6-step co-design process with children in a low socio-economic area of Amsterdam to create interventions for obesity prevention. Steps 1 and 2 in their process involved a needs assessment and creating community partnerships; steps that may have potentially benefited from a GMB process to promote community collaboration and participant engagement. Furthermore, we found that GMB alone was not sufficient to produce actionable interventions, and, therefore, a research process that incorporates both GMB and co-design may increase the ability of research to catalyse action within a community. 

Other researchers suggest creating a quantitative simulation model from the CLD created during the GMB process [15,28]. This approach checks the logical consistency of the community’s CLD using objective data. The resulting quantitative model presents stocks and flows which may also be more informative for policy decision making when, for example, an intervention could be shown to increase children’s vegetable consumption by 0.3 serves a day. In order to evaluate system changes after interventions were implemented, Brown (2019) repeated the GMB process with community participants to add additional variables and feedback loops. This could be a next step for the project [29].

## 5. Conclusions

GMB is an effective method for engaging participants in the process of building collective models to visualise a complex system. Underlying GMB is a worldview that sees behaviour arising out of the structure of a system. The system involves key variables such as ‘marketing of fast foods’ and a pattern of causal connections that, in this example, describe both the drivers of the marketing and its consequences. The focus of GMB is therefore on helping people develop an understanding of this system. In addition to understanding how to use specific GMB tools, participants learn to see causal connections and how these connections result in patterns of behaviour evolving over time. Any recommendations or action resulting from the research is therefore more likely to be successful.

This paper set out to explore three tools that collaboratively engaged community members in a GMB process to develop a rich understanding of the complexity of the food system and mobilise action on a public health issue. The three tools—graphs over time, cognitive mapping and causal loop diagrams—can be transferred to other public health research projects, to improve ways in which communities can be actively involved in investigating and taking action on complex issues. The study described in this paper was successful, as it resulted in a detailed description of the determinants of fruit and vegetable intake in children, and importantly, led to action within a community following an additional step of co-design led by the partnering community organisation. The GMB process described in this paper has built capacity within a community for systems thinking, which has led to a deeper understanding of public health and a desire among participants and community leaders to bring about positive change.

## Figures and Tables

**Figure 1 ijerph-17-03457-f001:**
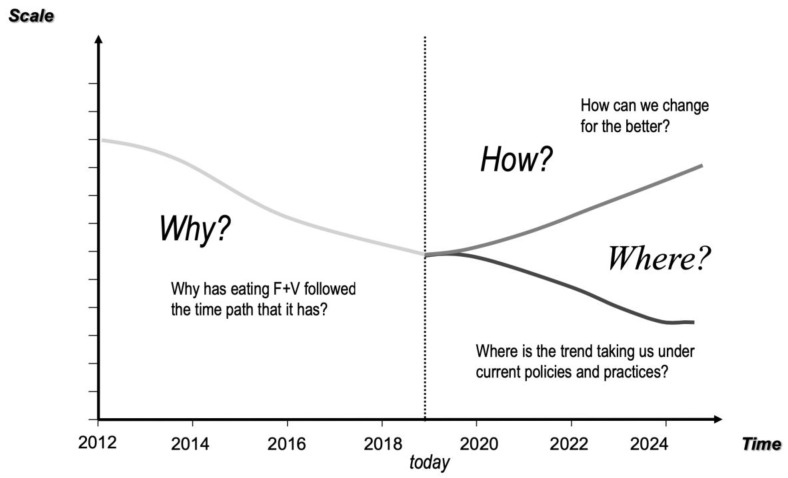
Graphs over time template.

**Figure 2 ijerph-17-03457-f002:**
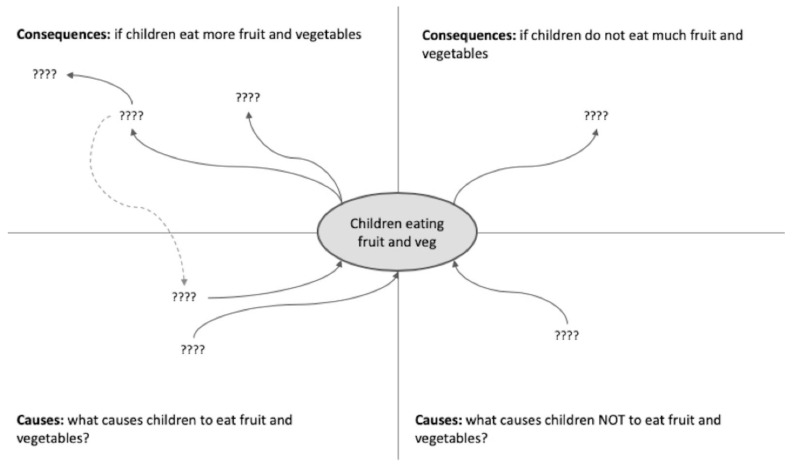
Causes and consequences template for cognitive mapping.

**Figure 3 ijerph-17-03457-f003:**
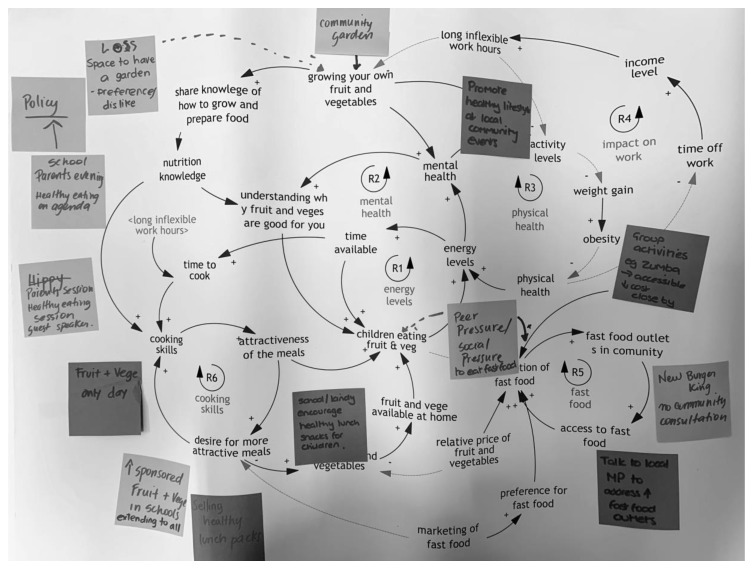
Example of actions placed on the collective causal loop diagram (CLD).

**Figure 4 ijerph-17-03457-f004:**
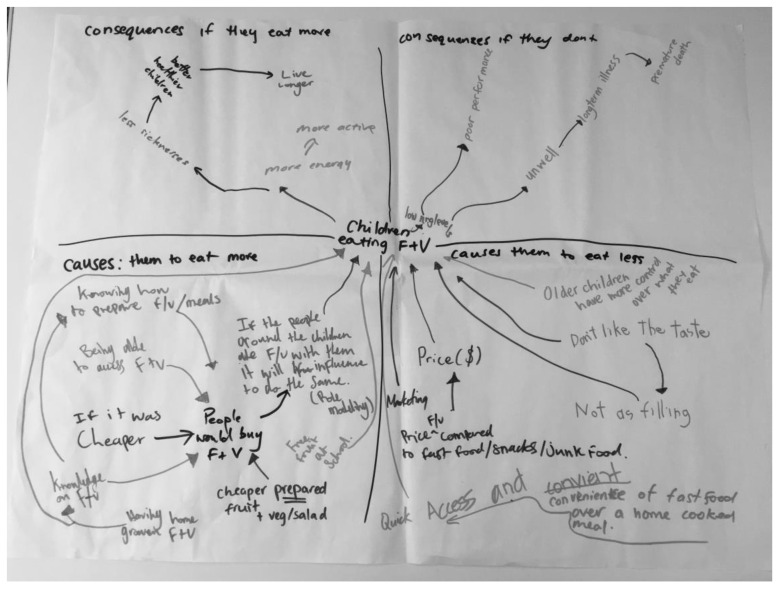
Example of completed causes and consequences template.

**Figure 5 ijerph-17-03457-f005:**
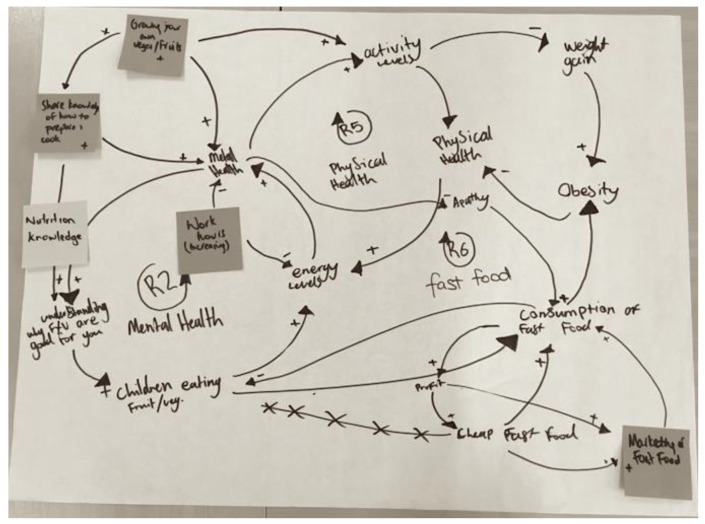
Example of early CLD from one small group.

**Table 1 ijerph-17-03457-t001:** Research team roles and tasks during the workshops.

Role	Tasks
Gatekeeper role (identifies and advocates for the participants, but is also aware of the needs of the team, the process and what needs to be achieved in the workshops)	Onsite 5.10 pm: help project coordinator to set up table for name tags/consent forms, welcoming participants as they arrive. Begin official welcome and karakia once everyone has arrived, approximately 5.30 pm (with kaumātua). Go through housekeeping: detail where the toilets and emergency meeting area are; phones off. Lead whakawhanaungatanga exercise/short icebreaker. During workshop: a ‘roamer’ moving around tables checking that everyone is happy and helping lead facilitator to hand out resources, etc., as needed. Taking photos during all three workshops.
“Lead facilitator” role (focused on the group dynamics)	In classroom before Workshop 1: setting up five group-work tables with materials, ensuring PowerPoint works, etc. Leading/closing sessions. Full group activities facilitator.
“Modeler” role (focused on the model that is being built)	In classroom before Workshop 1: setting up five group-work tables with materials, ensuring PowerPoint works, etc. Wall builder for graphs over time exercise. Short lecture on ‘thinking about systems’ and then lead the causes and consequences exercise.
Project coordinator	Main point of communication for participants, ensuring that they have preworkshop material, etc. Ensure that each participant has read the Participant Info Sheet and get them to sign consent form, give name tags. Check catering set up at back/side of classroom and water available, plates, cups, serviettes, etc. Table facilitator during workshop (check for understanding and that everyone is contributing, be alert for any problems, communicate with lead facilitator about how it is going, make sure the groups’ work is being recorded/captured in the written/modelling tasks).
Additional table facilitators (×4)	Welcoming participants as they arrive. Table facilitator during workshop (as above).

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
