# Peer review of "Community Group Model Building as a Method for Engaging Participants and Mobilising Action in Public Health"

_ijerph, 2020, doi:10.3390/ijerph17103457_

Round 1
Reviewer 1 Report
As I said in my previous review, I am supportive of this paper because it is an important perspective and approach in public health and it is rarely written about it sufficient detail that researchers new to the approach can learn from. The authors have done a good job to turn it more into a methods paper / how-to guide. I’ve got a few small suggestions as a follow up:
Introduction
Great to see the NZ study added; I’m not quite sure I understand the new sentence – a verb perhaps missing, and maybe a few shorter sentences could help with clarity? The main gist of their study was that they looked at the usefulness of a national model for local decision-makers and these stakeholders found it indeed very valuable but stressed that a national model would need to be adaptable to local contexts/data.
Methods/Results
As mentioned in my previous review, to make this into more of a ‘how-to guide’ paper, I’d present some of these reflections on the kind of data/insights you get from using these tools in your results not in the methods. I understand that you didn’t primarily set out to evaluate the usefulness of the tools but a few more tweaks might make it into a stronger methods paper. It just seems a bit unbalanced at the moment with a lengthy methods section and a short results section. As a suggestion, for 2.4 Tools, I’d just keep the general description for each tool and template in the methods, but then structure the results according to the same three tools and move anything on what the stakeholders did – in particular the examples for how they filled in the templates into the results section. I’d suggest that in a methods paper, the results cover all the operationalisation and evaluation of the methods.
Author Response
As I said in my previous review, I am supportive of this paper because it is an important perspective and approach in public health and it is rarely written about it sufficient detail that researchers new to the approach can learn from. The authors have done a good job to turn it more into a methods paper / how-to guide. I’ve got a few small suggestions as a follow up:
Introduction
Great to see the NZ study added; I’m not quite sure I understand the new sentence – a verb perhaps missing, and maybe a few shorter sentences could help with clarity? The main gist of their study was that they looked at the usefulness of a national model for local decision-makers and these stakeholders found it indeed very valuable but stressed that a national model would need to be adaptable to local contexts/data.
RESPONSE: Agree, we have amended to now read: “In New Zealand, one of the authors utilised a system dynamics simulation model to engage a wide range of stakeholders in exploring a range of options for cardiovascular disease interventions[9]. The Group Model Building (GMB) process that emerged out of this study emphasised that engaging key stakeholders in a model building process is important and an effective way to create collaborative models [10,11].”
Methods/Results
As mentioned in my previous review, to make this into more of a ‘how-to guide’ paper, I’d present some of these reflections on the kind of data/insights you get from using these tools in your results not in the methods. I understand that you didn’t primarily set out to evaluate the usefulness of the tools but a few more tweaks might make it into a stronger methods paper. It just seems a bit unbalanced at the moment with a lengthy methods section and a short results section. As a suggestion, for 2.4 Tools, I’d just keep the general description for each tool and template in the methods, but then structure the results according to the same three tools and move anything on what the stakeholders did – in particular the examples for how they filled in the templates into the results section. I’d suggest that in a methods paper, the results cover all the operationalisation and evaluation of the methods.
RESPONSE: Thank you, we agree. We have now moved some of the methods into the results section as suggested, and restructured the results section to reflect the format of the methods. We have also added more ‘how to’ information for readers around the roles of the research team (see new Table 1).
Reviewer 2 Report
The paper is much improved and is an interesting read. I could no see using this paper in a graduate methods class discussing different approaches to engaged research methods.
My only lingering concern is on lines 171-172. The authors say, "Learning how to build, read, and use CLDs creates a significant cognitive load on research participants." This sentence is unnecessarily jargony and the style is inconsistent with the rest of the paper and how the issue is discussed in the Discussion section on lines 261-263. The effect of the jargony wording on lines 171-172 is that the language reads as condescending towards the participants. Revise this sentence with more humanized language and the content is fine.
Thank you for your hard work on the revisions. I like this paper a lot.
Author Response
The paper is much improved and is an interesting read. I could now see using this paper in a graduate methods class discussing different approaches to engaged research methods.
My only lingering concern is on lines 171-172. The authors say, "Learning how to build, read, and use CLDs creates a significant cognitive load on research participants." This sentence is unnecessarily jargony and the style is inconsistent with the rest of the paper and how the issue is discussed in the Discussion section on lines 261-263. The effect of the jargony wording on lines 171-172 is that the language reads as condescending towards the participants. Revise this sentence with more humanized language and the content is fine.
Thank you for your hard work on the revisions. I like this paper a lot.
RESPONSE TO REVIEWER 2: Thank you for your encouraging comments and for picking up the jargon/condescension which certainly was not our intention. We agree, and have amended to “Some research participants can find CLDs challenging to build, read and use. Therefore…”
This manuscript is a resubmission of an earlier submission. The following is a list of the peer review reports and author responses from that submission.
Round 1
Reviewer 1 Report
This manuscript is very well written and sets out very clearly the steps for group model building as an engagement tool. I am largely supportive of this paper because it is an important perspective and approach in public health and it is rarely written about it sufficient detail that researchers new to the approach can learn from. However, I was disappointed that the actual results of the GMB were not shared within this article and are actually already published in a separate paper that shares the study outcomes in greater detail. This could work as a methods paper instead, but as a reader, it is difficult to judge the benefits and value of the method without some illustrative examples in what way this approach elicited discussions, new insights or understandings. Some selected products from the workshops are shown to illustrate three tools but left without description or commentary. Moving some of the content from the discussion into the results might help presenting it more as an evaluative or analytical paper – in particular the feedback from the community partners. As it stands for now, the paper is not a research paper but more of a commentary – which I’d be supportive of if the journal allows for such contributions.
Abstract
The abstract is well written. Reading it, I either expected a research paper that presents the findings of the study, or a methods paper evaluating the benefits and impacts of using this approach – as set out in these sentences of the abstract:
“GMB effectively engaged participants and increased their understanding of systems thinking. The resulting ‘systems map’ has been used to advocate for changes to the food environment and policies affecting children’s nutrition. Key learnings were the value of genuine partnership with a local public health programme for longevity of the project, recruitment of key decision-makers from the community early in the process, and allowing time to create sustainable change.”
I’d suggest making these points the core of the results section; and this would make it a great methods paper.
Introduction
The introduction describes the history of SDM and group model building well – and I think shows that it is not such a novel method but that it has received more recent recognition in public health as a participatory approach to engage participants and create impact beyond computer simulation models. A great New Zealand study seems missing that also explicitly asked involved stakeholders for their feedback – in their case of the usefulness of adapting a national SDM model to local settings/data: Kenealy 2012:36(6) Australian and New Zealand Journal of Public Health. Not explicitly a paper about GMB or nutrition but similar aims.
Methods
The methods are set out really well; few SDM papers describe group model building in sufficient detail to help other researchers undertake similar studies. The methods as presented here made me think that this is a research paper presenting the findings from this study – until I realised that these have already been published elsewhere. If this is more of a ‘how-to guide’ paper, this should be set out more clearly in the methods.
Results
This section was confusing because it only presented some example outputs without much explanation and reference to a paper that presents these findings. I’m not sure this therefore works as a stand alone paper – and if this is more of an how-to paper or methods evaluation, some content from the discussion should be in the results and it would require a more in-depth attempt to interrogate and analyse the value of using this method.
Discussion
See above.
Reviewer 2 Report
The paper presents an interesting method for engaging participants in constructing models about processes within their communities. Currently, there is not enough detail to follow what is happening in the paper but this can be addressed fairly reasonably.
1) Introduction: p1 line 41 - The authors should say what they mean by nutrition is embedded in a social, cultural, and political context. Provide examples to elaborate.
2) There needs to be more literature on CBPR and it should be clarified that GMB is not competing with CBPR as a framework. One could describe GMB as an approach that could be used within a CBPR approach. The authors imply that GMB is an equitable and fully distributed approach but this is not really demonstrated by the authors. There is no exposition on developing a connection with a community. It appears that the authors partnered with a CBO in order to carry out the research but there is no elaboration that suggests that this work comes out of the community. The methods are intended to get the community's perspective but the approach and research design itself appears to have come from the University. The authors should be more measured in their claims about the degree to which this is a CBPR approach and provide a better explanation of what CBPR is.
3) Were participants compensated? Was the study IRB approved?
4) Sections 2.4.1-2.4.3 should each provide an example of what the method is in the form of a generic diagram with an explanation of how the reader should interpret it.
5) The authors need to tell a story across Figures 2-5. What should I be getting out of each of these figures? How is what I am learning from participants developing across each of them? Currently, the figures are just plopped in the article. It would be helpful to show how the process evolved over the three workshops.